# Molecular Networking Leveraging the Secondary Metabolomes Space of *Halophila stipulaceae* (Forsk.) Aschers. and *Thalassia hemprichii* (Ehrenb. ex Solms) Asch. in Tandem with Their Chemosystematics and Antidiabetic Potentials

**DOI:** 10.3390/md19050279

**Published:** 2021-05-18

**Authors:** Nesrine M. Hegazi, Hamada H. Saad, Mona M. Marzouk, Mohamed F. Abdel Rahman, Mahitab H. El Bishbishy, Ahmed Zayed, Roland Ulber, Shahira M. Ezzat

**Affiliations:** 1Department of Phytochemistry and Plant Systematics, Division of Pharmaceutical Industries, National Research Centre, Dokki, Cairo 12622, Egypt; nm.hegazi@nrc.sci.eg (N.M.H.); mm.marzouk@nrc.sci.eg (M.M.M.); 2Department of Pharmaceutical Biology, Pharmaceutical Institute, Eberhard Karls University of Tübingen, 72074 Tübingen, Germany; 3Department of Biology and Biochemistry, School of Life and Medical Sciences, University of Hertfordshire Hosted by Global Academic Foundation, Cairo 72074, Egypt; m.farouk@gaf.edu.eg; 4Department of Pharmacognosy, Faculty of Pharmacy, October University for Modern Sciences and Arts (MSA), Giza 12451, Egypt; mahelmy@msa.eun.eg; 5Institute of Bioprocess Engineering, Technical University of Kaiserslautern, Gottlieb-Daimler-Straße 49, 67663 Kaiserslautern, Germany; ahmed.zayed1@pharm.tanta.edu.eg; 6Department of Pharmacognosy, College of Pharmacy, Tanta University, El-Guish Street (Medical Campus), Tanta 31527, Egypt; 7Department of Pharmacognosy, Faculty of Pharmacy, Cairo University, Kasr El-Aini Street, Cairo 11562, Egypt

**Keywords:** seagrasses, *Halophila stipulacea*, *Thalassia hemprichii*, Hydrocharitaceae, molecular networking, antidiabetic, chemosystematics

## Abstract

The Red Sea is one of the most biodiverse aquatic ecosystems. Notably, seagrasses possess a crucial ecological significance. Among them are the two taxa *Halophila stipulacea* (Forsk.) Aschers., and *Thalassia hemprichii* (Ehrenb. ex Solms) Asch., which were formally ranked together with the genus Enhalus in three separate families. Nevertheless, they have been recently classified as three subfamilies within Hydrocharitaceae. The interest of this study is to explore their metabolic profiles through ultra-high-performance liquid chromatography-high-resolution mass spectrometry (UPLC-HRMS/MS) analysis in synergism with molecular networking and to assess their chemosystematics relationship. A total of 144 metabolites were annotated, encompassing phenolic acids, flavonoids, terpenoids, and lipids. Furthermore, three new phenolic acids; methoxy benzoic acid-*O*-sulphate (**16**), *O*-caffeoyl-*O*-hydroxyl dimethoxy benzoyl tartaric acid (**26**), dimethoxy benzoic acid-*O*-sulphate (**30**), a new flavanone glycoside; hexahydroxy-monomethoxy flavanone-*O*-glucoside (**28**), and a new steviol glycoside; rebaudioside-*O*-acetate (**96**) were tentatively described. Additionally, the evaluation of the antidiabetic potential of both taxa displayed an inherited higher activity of *H. stipulaceae* in alleviating the oxidative stress and dyslipidemia associated with diabetes. Hence, the current research significantly suggested *Halophila*, *Thalassia*, and *Enhalus* categorization in three different taxonomic ranks based on their intergeneric and interspecific relationship among them and supported the consideration of seagrasses in natural antidiabetic studies.

## 1. Introduction

The Red Sea is remarkably regarded as one of the most biodiverse marine environments harbouring seagrasses with a pivotal ecological significance either as natural reservoirs of an array of microhabitats or perpetual promoters of countless adjacent living systems [1]. Taxonomically, seagrasses are belonging to five distinct families; Cymodoceaceae, Hydrocharitaceae, Posidoniaceae, Ruppiaceae, and Zosteraceae [2,3]. From which the Hydrocharitaceae family comprises 16 genera and 135 species; occurring mainly in freshwater habitats [4]. Nevertheless, only three genera of the family exist exceptionally in marine niches, and this includes *Halophila* Thouars (10 species), *Thalassia* Banks ex K.D.König (2 species), besides *Enhalus* Rich (monospecific) [5].

According to the Egyptian flora, these taxa are solely represented by two genera *Halophila* (two species; *Halophila stipulacea* (Forsk.) Aschers. and *Halophila ovalis* (R.Br.) Hook. f.), and *Thalassia* (one species; *Thalassia hemprichii* (Ehrenb. ex Solms) Asch.) [6,7]. Moreover, these species are not only distinguished from other Hydrocharitaceae members by their marine environment, but also, they share familiar morphological characters that differ from the rest of the species.

Despite their prevalence and environmental impact on their niches, limited phytochemical reports have unveiled the occurrence of unidentified sulphated phenolic compounds, non-sulphated flavones, flavone glycosides in *H. stipulacea* [8], and terpenoids exemplified by the macrocyclic glycoterpenoid syphonoside [9]. At the same time, acylated polyhydroxylated flavone glycosides were also found in *H. johnsonii* [10]. Within the same context, *Thalassia* depicted as *T. hemprichii* is known to afford benzoic acids, cinnamic acids, methoxy flavones, and sulphated flavone glycosides [11]. In contrast to the broad variability of the secondary metabolic content, the previous biological evaluations were merely narrowed to their antioxidant and antimicrobial potencies [8,11,12,13,14].

In the meantime, diabetes mellitus continually contributes to the underlying reason for several morbidities and has a myriad of complications, including cardiovascular disorders, blindness, kidney failure, and peripheral nerve damages, and thus represents a significant threat to global health [15]. To partially sort out such implications, several hypoglycemic strategies have been recruited for the management of diabetes, including insulin release stimulation, gluconeogenesis inhibition, glucose transport activity increase, and intestinal glucose absorption reduction [16]. Nonetheless, the World Health Organization (WHO) Expert Committee constantly urges the need to come up with new hypoglycemic agents from natural origins to lessen the side effects of the marketed synthetic drugs [17]. In response, our phytochemical traction was channeled to the marine environment, particularly seagrasses, as a relatively underexploited source of bioactive metabolites possessing engaging and less investigated pharmacological aspects [18].

Motivated by the abundance and phytochemical richness of such species besides the pressing need of finding alternative hypoglycemic sources, our contemporary quest aims to broadly characterize the metabolic profiles of both *H. stipulacea* (*Hs*) and *T. hemprichii* (*Th*) through a comprehensive ultraperformance liquid chromatography–high-resolution mass spectrometry (UPLC–HRMS-MS) analysis supported by the Global Natural Products Social Molecular Networking (GNPS) and to shed light on their chemosystematics relatedness with other seagrass species. To correlate the possible antidiabetic potentiality in tandem with the annotated chemistries, in vitro inhibitory assessments against different digestive enzymes and in vivo antidiabetic assays were conducted as well. It may help the discovery of novel marine-derived drugs from one of the richest, biodiverse, and less exploited environments and open new frontiers revealing the metabolic profiles responsible for such activity.

## 2. Results and Discussion

### 2.1. Phytochemical Analysis

#### 2.1.1. Acid Hydrolysis

Complete acid hydrolysis of both *H. stipulacea* (*Hs*) and *T. hemprichii* (*Th*) extracts was performed for the sake of the isolation of the major aglycones and the identification of the linked sugar moieties. The pure aglycones were isolated from the ethyl acetate layers of *Hs* and *Th* and gave dark spots on PC under UV, indicating the flavone nucleoli. Comparing with authentic samples, some spots showed matching *R_f_* values and color reaction as that of apigenin, genkwanin, scutellarein, isoscutellarein, hispidulin, luteolin, chrysoeriol, 6-hydroxyl luteolin, and pedalitin. Furthermore, the major detected aglycones (apigenin, isoscutellarein, cirsimaritin, luteolin, chrysoeriol, and 6-hydroxyl luteolin) were additionally confirmed via their UV and ^1^H-NMR spectral data (Appendix A). Glucose was spotted as a significant sugar moiety in the aqueous extract in addition to xylose and rhamnose moieties. Complete hydrolysis confirmed that all glycosides were in *O*-glycosidic format.

#### 2.1.2. High-Performance Liquid Chromatography (HPLC) and Ultra-High Performance Liquid Chromatography-High-Resolution Mass Spectrometry (UPLC-HRMS/MS) Analyses

The HPLC readings of *Hs* and *Th* extracts at different wavelengths (Appendix A) exhibited significantly different profiles characterized by two dominant peaks at 26 min in *Hs*, unlike the *Th* profile, which disclosed more UV detectable metabolites chiefly eluted from 19–28 min.

Consequently, a comprehensive analysis of the metabolites of the two seagrasses was commenced using reversed-phase (RP-C_18_) ultra-performance liquid chromatography (UPLC) coupled to photodiode array detection (PDA) and electrospray ionization (ESI) tandem mass spectrometry, i.e., UPLC-PDA-ESI-HRMS/MS in both positive and negative ionization modes for the in-depth exploration of the metabolic differences between the two species (Appendix A).

#### 2.1.3. UPLC-HRMS/MS Metabolite Annotation Aided with Molecular Networking

The UPLC-MS/MS data were mined employing GNPS platform (Global Natural Products Social Molecular Networking) in which molecular networking (MN) was generated to visually display the existing chemical space of the acquired MS/MS data and infer the metabolites distribution facilitating the dereplication across different metabolomes [19].

Two MNs were laid out from the MS/MS data of both ionization modes. The negative MN resulted in 314 nodes grouped as 19 clusters (with a minimum of two connected nodes) and 190 singletons (Figure 1). The significant dereplicated sets of the negative MN were cluster **A** (lipids), cluster **B** (flavonoids), cluster **C** (cinnamic acid esters), cluster **D** (steviol glycosides), and cluster **E** (macrocyclic glycoterpenoids) (see Figure 1).

In parallel, the positive MN constituted of 217 nodes in 18 clusters and 58 discrete nodes, in which cluster **A** (flavonoid glycosides and their aglycones) and cluster **B** (cinnamic acid amides) were of interest (Appendix A). In general, nodes were portrayed as a pie chart to reflect the relative abundance of each ion in the two extracts. The two samples were color-coded, where purple represents *Hs* extract and yellow as *Th* extract. In contrast, the solvent as a blank was given a black color to be avoided during the annotation phase.

Metabolites putative annotation was fundamentally counted on considering their retention times, chemical formula, UV absorption maxima, along with their fragmentation behavior with hitherto described literature. The propagation of metabolites identification was further reinforced with the MN inspection, and GNPS spectral library search synchronized with the proposed fragmentation trees by Sirius [20,21]. Overall, 144 metabolites were tentatively assigned, encompassing phenolic acid derivatives (35 metabolites), flavonoids (48 metabolites), terpenoids (5 metabolites), lipids (11 fatty acids, 13 acylglycerol, and 9 phospholipids, and others. Detailed information about the annotated metabolites (including the retention time, observed molecular ion, fragmentation pattern, molecular formula, and mass error) are displayed in Appendix A. Among them, annotated chemotypes like phenolic acids, flavonoids, and terpenoids were exploited as useful chemotaxonomic markers (Figure 2a,b).

##### Phenolic Acids

Two distinct subgroups of phenolic acids were observed, including benzoic and cinnamic acid derivatives, which were more characteristic of *Th* extract. Generally, neutral losses of 44 Da stands for the cleavage of the carboxylic group of the phenolic acids, while the expulsions of 80 and 162 Da were typical for sulphated and glucosylated additives, respectively.

##### Benzoic Acids

Earlier, benzoic acids were detected in *Th* and some *Halophila* species based on comparative chromatography [22] and were later isolated from *Th* to include *p*-hydroxyl benzoic acid [23]. Benzoic acids were found to be either glucosylated or sulphated in the negative MN as isolated nodes, which could be justified by their fewer observed fragments (2–4) vs. the used minimum number of common fragments (set to 6) required to cluster two spectra in the MN. This to include two isomers of vanillic acid-*O*-glucoside; **8** and **13** (*m/z* 329.0874 [M − H]^−^, C_14_H_18_O_9_), syringic acid-*O*-glucoside; **11** (*m/z* 359.0982 [M − H]^−^, C_15_H_20_O_10_), in addition to a benzaldehyde derivative annotated as vanillin-*O*-glucoside; **10** (*m/z* 313.0927 [M − H]^−^, C_14_H_18_O_8_); all sharing the loss of a glucoside moiety as apparent from a fragment ion at *m/z* [M-H-162]^−^.

Furthermore, a new sulphated benzoic acid derivative **16**, an isolated node in the negative MN found in both extracts, was tentatively detected at Rt 11.4 min with C_8_H_8_O_6_S, *m/z* 230.9970 [M − H]^−^ proven by the observed fragment ions at *m/z* 151 [M-H-80]^−^, and 107 [M-H-80-44]^−^ (Appendix A). Thus, it was assigned as methoxy benzoic acid -*O*-sulphate. In a similar fashion, an exclusive feature **30** to *Th* extract at 13.25 min with an extra 30 Da (OCH_2_) was tentatively envisioned as a dimethoxy benzoic acid-*O*-sulphate (*m/z* 261.0075 [M − H]^−^, C_9_H_10_O_7_S) considering its fragments at *m/z* 181 [M-H-80]^−^ and 137 [M-H-80-44]^−^ (Appendix A).

Within the same chemical space, a diglycosylated benzoic acid congener was also proposed as a dihydroxybenzoylmethyl ester-(-*O*-vanilloyl)-xylosyl glucoside; **44** (*m/z* 611.1619 [M − H]^−^, C_27_H_32_O_16_) based on its MS^2^ spectrum which gave *m/z* 461 [M-H-C_8_H_8_O_3_]^−^ pointing to the cleavage of a vanilloyl motif, and fragments at *m/z* 311 and 167 corresponding to the breakup of the xylosyl-glucoside units followed by the dihydroxybenzoyl methyl ester moiety.

##### Cinnamic Acids and Their Derivatives

An extra set of phenolic acids was deconvoluted as cinnamic acid-containing compounds, chiefly derivatives of caffeic acid; **39** (*m/z* 179.0351 [M-H], C8H8O4), coumaric acid; **64** (*m/z* 163.0402 [M − H]^−^, C_9_H_8_O_3_), and ferulic acid; **93** (*m/z* 193.0505 [M − H]^−^, C_10_H_10_O_4_).

Similar to the earlier discussed sulphated benzoic acids, two variants of sulphated cinnamic acid were also detectable exemplified by hydroxyphenyl-sulfooxypropanoic acid (tichocarpol A); **12** in *Hs* (*m/z* 261.0072 [M − H]^−^, C_9_H_10_O_7_S), previously isolated from red marine algae *Tichocarpus crinitusin* with potent feeding-deterrent activity against sea urchins [24]. Besides, **34** in *Th* (*m/z* 258.9919 [M − H]^−^, C_9_H_8_O_7_S) was assigned as sulphated caffeic acid obeying the characteristic fragmentation mechanism of the sequential cleavage of sulphate (−80 Da) and carboxylic groups (−44 Da).

Equivalently to benzoic acid glucosides dispersal in MN, the glucosylated formats of cinnamic acid were visualized too as scattered nodes in the negative MN owing to their few detected fragments, however, still marked by the neutral loss of 162 Da. They were initially recorded as sinapic acid-*O*-glucoside; **19** (*m/z* 385.1138 [M − H]^−^, C_17_H_22_O_10_) followed by coumaric acid-*O*-glucoside isomers; **20** and **24** with EIC at Rt 12.33 and 12.81 min (*m/z* 325.0923 [M − H]^−^, C_15_H_18_O_8_). Equally, isomers of dihydrocoumaroyl-*O*-glucoside were additionally seen at Rt 12.49 and 12.57 min, **21** and **23** (*m/z* 327.1083 [M − H]^−^, C15H20O8), and lastly ferulic acid-*O*-glucoside; **31** (*m/z* 355.1035 [M − H]^−^, C_16_H_20_O_9_).

Moreover, a suite of mono- and di-substituted tartaric acid esters was disclosed seamlessly in *Th* sample as **17**, **26**, **32**, **57**, **58**, **74**, **78**, **80**, **81**, **91**, and **95** composing cluster **C** in the negative MN (Figure 1). They were deciphered by 132 Da as a neutral loss accounting for the breakage of dehydrated tartaric acid (C_4_H_4_O_5_) and the distinct fragment at *m/z* 149, C_4_H_6_O_6_ for the tartaric acid moiety.

Monocinnamic acid esters of tartaric acid including *O*-caffeoyl tartaric acid isomers; at Rt 11.6 and 17.38 min; **17** and **57** (*m/z* 311.0439 [M − H]^−^, C_13_H_12_O_9_), *O*-coumaroyl tartaric acid; **32** (*m/z* 295.0460 [M − H]^−^, C_13_H_12_O_8_) and *O*-feruloyl tartaric acid; **81** (*m/z* 325.0565 [M − H]^−^, C_17_H_14_O_6_) were laid out as standalone ions in the negative MN thanks to the recruited parameters.

Nevertheless, di-substituted tartaric acid entities were grouped efficiently in the negative MN as cluster **C** and were distinctive of *Th* (Figure 1). This listed di-*O*-caffeoyl tartaric acid; **58** (*m/z* 473.0724 [M − H]^−^, C_22_H_18_O_12_) which showed distinctive fragments at *m/z* 311 and 149 for the consecutive losses of two caffeoyl acid moieties, followed by *O*-caffeoyl *O*-feruloyl tartaric acid; **80** (*m/z* 487.0880 [M − H]^−^, C_23_H_20_O_12_) with fragments at *m/z* 325 and 149 for the loss of a caffeoyl and feruloyl moieties, respectively.

Remarkably, isomeric di-substituted tartaric acid esters were well-differentiated within cluster **C** in the negative MN (Figure 1) through their retention time discrepancy. For instance, isomers of *O*-caffeoyl-*O*-coumaroyl tartaric acid; **74** and **78** at Rt 19.25 and 19.59 min, (*m/z* 457.0777 [M − H]^−^, C_22_H_18_O_11_) gave fragments at *m/z* 295 and 149 for the cleavage of a caffeoyl and coumaroyl units, respectively. Likewise, di-*O*-coumaroyl tartaric acid isomers; **91** and **95** at Rt 21.29 and 21.70 (*m/z* 441.0828 [M − H]^−^, C_22_H_18_O_10_) with a base peak at *m/z* 163 corresponding to the coumaric acid moiety.

Ultimately, a newly non-reported congener of caffeoyl tartaric acid ester, **26** (*m/z* 491.0829 [M − H]^−^, C_22_H_20_O_13_), observed as a singleton in the negative MN, was tentatively possible to be read out in terms of structural anticipation. Its suggestive skeleton was proposed with the aid of fragments observation at *m/z* 329 [M-H-162]^−^, 179 [M-H-329-150]^−^ for the cleavage of caffeic and tartaric acid motifs, followed by the loss of two CH_3_ groups (*m/z* 149) assisting its dereplication as *O*-caffeoyl *O*-hydroxyldimethoxybenzoyl tartaric acid (Appendix A).

Despite the variation in the chemical scope of the monitored cinnamic acid esters in *Th,* two isomers of cinnamic amides; **38** and **43** were utterly identified in *Hs* by the positive MN at Rt 14.7 and 15.37 min; as dicoumaroyl spermidines (*m/z* 438.2405 [M + H]^+^, C_25_H_31_N_3_O_4_) (Appendix A).

The occurrence of cinnamic acids was previously described in *Thalassia* [25], *Halophila* [26], and various seagrass species [27]. Similarly, tartaric acid esters of cinnamic acids were previously encountered in *Thalassia* [23] and some seagrass taxa such as *Syringodium* [28], *Cymodocea* [29], and *Posidonae* [30] mainly as caftaric and chicoric acids.

##### Flavonoids

Recently, there is a growing interest in marine-derived flavonoids, which may be attributed not only to their auspicious biological potential, but also for their antifouling and anti-feedant activities. The reported biological activities of marine flavonoids exhibited a broad spectrum of potencies ranging from antioxidant, antimicrobial, antidiabetic, antitumor, anticoagulant, to antidiabetic [31].

Considering the seagrasses, upon investigation the flavone-type flavonoids were the most abundant class in both extracts, delivered as cluster **B** (Figure 1) in the negative MN. The fragmentation sequence was typical of *O*-glycosidic derivatives, being characterized by neutral losses and mass differences in the MN of 162, 146, or 132 correlating to *O*-hexoside, *O*-deoxyhexoside or *O*-pentoside, correspondingly [16]. The aglycone and sugar identities were confirmed as formerly stated via the acid hydrolysis of the two extracts to figure out the prevalence of the flavone aglycones and glucose as the main hexose, rhamnose as the deoxyhexoside, and xylose as the predominant pentose (Appendix A).

Flavone di-glucosides existed exclusively in *Hs* supported by the positive MN (Appendix A). They were annotated as apigenin di-*O*-glucoside **27** (*m/z* 595.1672 [M + H]^+^, C_27_H_30_O_15_) and chrysoeriol di-*O*-glucoside **29** (*m/z* 625.1785 [M + H]^+^, C_28_H_32_O_16_) evident by the consecutive losses of two glucose moieties.

Whereas flavone mono-glucosides were detected in both extracts, nevertheless they showed some differences in the aglycone moieties. For example, methoxylated flavone glucosides were distinctive of *Hs* and were witnessed in the positive MN (Appendix A) as three isomers of dihydroxy dimethoxy flavone-*O*-glucoside; **36**, **84**, **99** (*m/z* 477.1404 [M + H]^+^, C_23_H_24_O_11_), monohydroxy trimethoxy flavone-*O*-glucoside; **62** (*m/z* 491.1555 [M + H]^+^, C_24_H_26_O_11_). Besides, (rhamnosyl)-ethenyl-trihydroxyflavone (drymariatin A); 25 occurred in *Hs* extract (*m/z* 459.1299 [M + H]^+^, C_23_H_22_O_10_) with a characteristic fragment at *m/z* 283 for the cleavage of the ethenyl side-chain together with the rhamnosyl unit.

However, polyhydroxylated flavone glucosides were predominant in Th as unveiled from the negative MN (Figure 1), which could be exemplified by two isomers of 6-hydroxy luteolin *O*-glucoside; **37** and **48** (*m/z* 463.0878 [M − H]^−^, C_21_H_20_O_12_), 6-hydroxy luteolin *O*-rutinoside; **42** (*m/z* 609.1463 [M − H]^−^, C_27_H_30_O_16_), 6-hydroxy luteolin-*O*-xyloside; **56** (*m/z* 433.0776 [M − H]^−^, C_20_H_18_O_11_). Isoscutellarein 7-*O*-xyloside; **67** (*m/z* 417.0828 [M − H]^−^, C_20_H_18_O_10_), and isoscutellarein 7-O-glucoside; **69** (*m/z* 447.0931 [M − H]^−^, C_21_H_20_O_11_) were also observed, while scutellarein-*O*-glucoside; **61** (*m/z* 449.1086 [M + H]^+^, C_21_H_20_O_11_) was eluted at earlier retention time and characteristic for *Th* extract.

Additionally, both MNs indisputably enabled the retrieval of some common flavone monoglucosides from both extracts (Figure 1) (Appendix A). This to include pedalitin-*O*-glucoside; **51** (*m/z* 477.1038 [M − H]^−^, C_22_H_22_O_12_), apigenin 7-*O*-glucoside; **63** (*m/z* 431.0988 [M − H]^−^, 433.1135 [M + H]^+^, C_21_H_20_O_10_), chrysoeriol-*O*-glucoside isomers; **66** and **94** (*m/z* 461.1095 [M − H]^−^, 463.1247 [M + H]^+^, C_22_H_22_O_11_), and genkwanin 4′-*O*-glucoside; **97** (*m/z* 491.1281 [M ‒ H + FA]^−^, 447.1295 [M + H]^+^, C_22_H_22_O_10_).

Aside from the abundant flavone type glucosides, three flavanone type glucosides were also distinguished. The first **28** was noticed at Rt 13.41 min (*m/z* 513.1254 [M + H]^+^, C_22_H_24_O_14_), exclusively in Hs as a distant self-looped node. Its MS^2^ spectrum showed the loss of a glucoside group (-162 Da) with fragment at *m/z* 351 and *m/z* 331 for further dehydration (-18 Da). Consequently, it was tentatively dereplicated as non-previously reported methoxypentahydroxyflavanone-*O*-glucoside (Appendix A), whereas **33** (*m/z* 449.1090 [M − H]^−^, C_21_H_22_O_11_) and **52** (*m/z* 465.1036 [M − H]^−^, C_21_H_22_O_12_), both in *Th*, were putatively interpreted as tetrahydroxy flavanone-*O*-glucoside and pentahydroxy flavanone-*O*-glucoside, respectively.

##### Sulphated Flavonoids

Cluster **B** in the negative MN allowed the expansion of the flavonoids chemical space by instant recognition of sulphated motifs via 80 Da mass difference as an edge connection to their flavonoid glycosides (Figure 1) innate to *Th*. This agrees well with the previous investigations, which hinted at the occurrence of sulphated flavone glucosides in *Th* exemplified by Thalassiolins A-D, which showed antiretroviral activity via the inhibition of HIV integrase [32,33]. The successfully deciphered sulphated flavonoids included thalassiolin A; luteolin *O*-glucoside sulphate; **46** (*m/z* 447.0501 [M − H]^−^, 529.0662, [M + H]^+^, C_21_H_20_O_14_S) besides, thalassiolin B; chrysoeriol 7-*O*-glucoside sulphate; **59** (*m/z* 541.0635 [M − H]^−^, 543.0816 [M + H]^+^, C_22_H_22_O_14_S), thalassiolin C; apigenin7-*O*-glucoside sulphate; **60** (*m/z* 511.0550 [M − H]^−^, C_21_H_20_O_13_S), and pedalitin-*O*-glucoside sulphate; **75** (*m/z* 557.0601 [M − H]^−^, C_22_H_22_O_15_S).

Supplemental flavonoid sulphates were possible to be putatively defined as luteolin *O*-glucoside sulphate sodium salt; **45** (*m/z* 549.0313 [M − H]^−^, C_21_H_19_NaO_14_S), and luteolin-*O*-sulphate; **86** (*m/z* 364.9965 [M − H]^−^, C_15_H_10_O_9_S) in the form of singletons in the negative MN.

##### Acylated Flavonoids

In addition to the previously annotated sulphated, mono, and di-glycosylated flavonoids, acylated ones were decoded in both extracts as well. Acylated flavonoids in *Hs* extract were represented by acetylated and malonylated variants recognized as cluster **A** in the positive MN showing edge connectivity with mass shift of 42 and 86 Da, respectively (Appendix A) [15]. Malonylated derivatives were formerly defined in *Hs* by [34] and were encountered in this study comprising *O*-malonylated glucosides of 6-hydroxylluteolin; **54** (*m/z* 549.0879 [M − H]^−^, 551.1052 [M + H]^+^, C_24_H_22_O_15_), apigenin; **76** (*m/z* 519.1144 [M + H]^+^, C_24_H_22_O_13_), and chrysoeriol; **79** (*m/z* 549.1252 [M + H]^+^, C_25_H_24_O_14_). While, *O*-acetylated glucosides were symbolized by those of 6-hydroxy luteolin; **70** (*m/z* 505.0976 [M − H]^−^, 507.1147 [M + H]^+^,C_23_H_22_O_1_3) and apigenin; **89** (*m/z* 473.1088 [M − H]^−^, 475.1249 [M + H]^+^,C_23_H_22_O_11_).

Resembling the malonylated derivatives in *Hs*, coumaroyl flavone glucosides were merely noticed in *Th* extract as verified by the negative MN (Figure 1) and their fragmentation pattern showed a neutral loss of 146 Da (dehydrated coumaroyl moiety). The MN disclosed the co-existence of 6-hydroxyl luteolin-O- coumaroyl glucoside isomers; **83** and **87** and confirmed through their EIC at Rt 20.22 and 20.89 min; (*m/z* 609.1247 [M − H]^−^, C_30_H_26_O_14_) along with scutellarein-*O*-coumaroyl glucoside; **88** (*m/z* 593.1300 [M-H], C_30_H_26_O_13_).

##### Flavonoid Aglycones

Thirteen flavonoid aglycones were concluded during this study and united nicely in cluster **A** of the positive MN (Appendix A), nine of which were isolated and confirmed through their chromatographic and spectral behavior as earlier described (Appendix A).

##### Diterpenoids

Besides leveraging the chemical landscape of the phenolic acids and flavonoids entities, the negative MN was additionally able to unfold an extra phytochemical molecular family, diterpenoids, which were subdivided into two main classes: steviol glycosides and macrocyclic diterpenoids.

##### Steviol Glycosides

Steviol glycosides are diterpene glycosides well known for their sweetness and occur only in few plants, mainly *Stevia rebaudiana* [35]. They were reported to exert antioxidant, glucose-lowering, and antihypertensive effects [35]. During this study, three steviol glycosides were only detected in *Hs* extract and were unveiled based on their previously reported data in the literature. They were observed in the negative mode MN (Figure 1), firstly as rebaudioside B; **72** (*m/z* 803.3714 [M − H]^−^, C_38_H_60_O_18_) with fragmentation sequence showing the sequential loss of three hexoses moieties with fragments at *m/z* 641, 479, and 317 [36]. It was found to be connected to another analogue with a mass difference of 42 Da (C_2_H_2_O), implying an acetylated derivative **96** (*m/z* 845.3822, C_40_H_62_O_19_). Even though it shared the same aglycone as **72** at *m/z* 317, its fragmentation pattern was a bit different since it schemed an initial loss of two hexose units with fragments at *m/z* 683, 521 followed by a fragment at *m/z* 317 for the loss of an acetylated hexose. Thus, it was assigned as a new acetylated equivalent of rebaudioside B (Appendix A).

Lastly, as an observed scattered node in the negative MN rubusoside was characterized; **100** (*m/z* 641.3191 [M − H]^−^, C_32_H_50_O_13_) with a fragmentation order showing the cleavage of two hexose units yielding the aglycone at *m/z* 317 [37]. The declustering behavior from its similar entities (Figure 1) could be possibly envisioned to their differences in the position and degree of glycosylation.

##### Macrocyclic Glycoterpenoids

The existence of macrocyclic glycoterpenoids was formerly pinpointed in *Hs* and showed to suppress apoptosis in some human and murine cancer cell lines [9]. Nonetheless, the MN revealed their presence in *Th* as well for the first time (Figure 1, Appendix A). The detected glycoterpenoids comprised syphonoside; **53** (*m/z* 831.3645 [M+CH_2_O_2_-H]^−^, C_38_H_58_O_17_), and its acetylated derivative; **73** (*m/z* 873.3759 [M+CH_2_O_2_-H]^−^, C_40_H_60_O_18_) which were furtherly validated through their direct linkage in the negative MN (Figure 1) with Δ *m/z* of 42 Da (C_2_H_2_O) (Appendix A).

### 2.2. Chemosystematic Significance

At the intergeneric point of view, *Halophila*, *Thalassia* and *Enhalus* provide a remarkable example of intolerance concerning the phylogenetic similarities of seagrasses, even those presumed to be closely related. Formerly, refs. [38,39] assigned them to three separate families Thalassiaceae, Halophilaceae, and Vallisneriaceae, respectively, based on the morphological characters of their leaves and anthers. Nevertheless, they were later replaced within Hydrocharitaceae family separated amongst three different subfamilies Halophiloideae, Thalassioideae, and Vallisnerioideae, respectively [40,41]. This was further supported by the rbcL gene sequence [40], which deduced that these genera form a monophyletic group with an affinity for tropical oceanic habitats. Yet, this clade is imbedded within Hydrocharitaceae freshwater genera with 100% bootstrap support. Accordingly, the three marine genera were retained as a single taxon (e.g., a common subfamily) within the Hydrocharitaceae rather than as three distinct subfamilies of Hydrocharitaceae or three different seagrass families. Likewise, it was suggested that the family rank is not necessary for these genera; and the subfamily position within the Hydrocharitaceae can express a satisfactory classification [2].

Conspicuously, these genera are not only distinguished from other members of Hydrocharitaceae by their aquatic habitat but also, they share familiar morphological characters that differ from the rest of the aquatic species. For instance, their pollen grains are carried in chains, as threads of drips, while they are free and globular in the other Hydrocharitaceae species. These pieces of evidence support their discrepancy from other Hydrocharitaceae members and their significant affinity to seagrasses.

Yet, no prior chemosystematic studies at the intergeneric level were drafted for the marine Hydrocharitaceae; thus, the inquiry has persisted whether these marine genera represent one, two, or three independent origins within the Hydrocharitaceae family or as three distinct families.

Scrutinizing the phytochemical data obtained in the present study and in relation to previous ones, the phenolic profiles of the three genera were proven to be exploited as useful chemotaxonomic markers, both at the inter-specific and intra-specific levels (Figure 3).

Notably, the three marine genera (*Enhalus*, *Halophila*, and *Thalassia*) can biosynthesize sulphated derivatives of benzoic acids, flavones as well as cinnamic acid esters (Appendix A), some of which were previously detected in other seagrass taxa [28], and not reported in some possible freshwater genera of Hydrocharitaceae to the best of our knowledge.

The MNs (Figure 1 and Appendix A) showed the characterization of the Egyptian *Th* by the presence of phenylpropanoic acid esters and cinnamic acid derivatives of tartaric acid (Figure 1, cluster **C**). Particularly, *Th* was described with sulfated flavone glycosides (Appendix A, Figure 3), which is compatible with the previous chemical investigations on *Th* and *T. testudinum* [23] versus one sulphated flavone aglycone detected in both species which again fortifies the fact that the earlier species are considered to be twin species. Another distinguishable feature was the *O*-glycosylation of the flavone aglycones with xylose and glucuronioic acid in *Thalassia* and *Enhalus*, respectively [11,42].

Alternately, flavone acyl glycosides (as acetylated and malonylated variants), methoxylated flavanones, and steviol glycosides were exclusive to *Halophila* versus their abolishment in *Thalassia*. Similarly, methoxy flavones are restricted to some *Halophila* species, Th and *T. testudinum*, and are absent in *Enhalus acoroides* (L. f) Royle [42]. These metabolic differences supported their placement in different taxonomic ranks. Contrariwise, the freshwater taxa of Hydrocharitaceae are able to afford flavonol nuclei, and chlorogenic acid isomers [43,44], which are absent in the three marine genera.

Decisively, the phenolic entities, i.e., the extensive varieties as derivatives of benzoic acids, cinnamic acid esters, and flavonoids, of *Enhalus*, *Halophila*, and *Thalassia* could be efficient chemotaxonomic markers advocating their placement in three different taxonomic ranks. Nonetheless, the query of their affinity to three individual seagrass families or as three subfamilies of the Hydrocharitaceae remains unanswered, taking into regard that the former molecular biology study was based solely on one gene sequence. Additional phytochemical efforts (using UPLC-HRMS/MS analysis and MN) coupled with phylogenetic analysis (using other gene sequences) are required to compare them to other freshwater taxa of the Hydrocharitaceae to solve the existing taxonomic conflict.

### 2.3. In Vitro and In Vivo Antidiabetic Assays

Considering the growing declarations of diverse plants-based phytochemicals ability to control hyperglycemia through the suppression of the digestive enzymes and hence reducing the absorption of sugars and fatty acids [2,41,45]. It was of interest to test the marine-derived specialized metabolites, particularly seagrasses under investigation *Hs* and *Th*, for their antidiabetic potential via their in vitro inhibitory effects on some digestive enzymes, like α-amylase, *β*-glucosidase, and pancreatic lipase enzymes.

Both seagrasses showed comparable enzymatic inhibitions, highlighting their potential as antidiabetic agents to surveil the calorie intake (Appendix A). These findings were consistent with the previous dose-dependent inhibition results attained with *H. beccarii* [46] and *Th* [47].

Such observed inhibitory readings could possibly be rationalized to the high prevalence of phenolic metabolite(s) in both seagrasses (Appendix A), which reportedly regulate the carbohydrates and lipids metabolism by downregulating the digestive enzymes [48,49].

To gain more mechanistic insights regarding the antidiabetic effects of *Hs* and *Th*, the in vivo studies were recalled. Animals with high serum glucose and low serum insulin levels were considered, such as diabetic rats, and were included in the study. Interestingly, administration of dose-dependent *Hs* extract resulted in a glucose-lowering action and restored back the insulin levels (Figure 4a,b and Appendix A), which can be referred to its exclusive content of steviol glycosides [50]. Since diabetes is usually allied with pathophysiological conditions attributed to the dysregulation of the glucose transporter GLUT2 [51], the evaluation of GLUT2 levels was of interest. Despite the fact that both extracts were able to reverse the observed low GLUT2, the alleviating effect of *Hs* extract was nearly doubled either with the low or high administered dosage compared to *Th* (Figure 4c and Appendix A).

In alignment with the formerly attained glucose/insulin effects, the unique occurrence of steviol glycosides in *Hs* extracts possibly enabled the upregulation of GLUT2 gene expression, improving the uptake of glucose in the liver, which will in turn aid to ameliorate hyperglycemic conditions as formerly sketched [52,53,54,55,56].

Minding the controversial levels of serum nitric oxide (NO) exerted by its diverse roles played in the body ranging from maintaining the endothelial function to acting as an inflammatory mediator, the NO oxide levels were additionally listed [57,58]. The analyses iteratively revealed *Hs* extracts delivered 3–4 folds increase in the NO levels versus *Th* (Figure 4d, Appendix A), which could be due to the predominant phenolics and steviol glycosides [50].

To estimate the imbalance between oxidant/antioxidant status, which normally associates diabetes, malondialdehyde (MDA), the product of lipid breakdown, was selected as a valuable indicator of free radical-induced lipid peroxidation. Predictably counting on the phenolic content richness [59], both extracts were found to significantly decrease the pancreatic MDA, where *Hs* extracts showed the most notable antioxidant behavior (Figure 4e, Appendix A).

Abnormal lipid metabolism as an additional common diabetes complication linked to several cardiovascular disorders was also regarded within the in vitro tests’ suite [60]. The antihyperlipidemic effect was noticeable with both extracts altering the serum lipid profile to normality, highlighting the superiority of *Hs* extract as a lipid-lowering agent.

## 3. Materials and Methods

### 3.1. Plant Material

One kilogram of both *H. stipulacea* (*Hs*) and *T. hemprichii* (*Th*) were collected from the Red Sea of Ras Shetan Nuweiba in Egypt. A voucher specimen (17-1-2018 for *Hs* and 18-1-2018 for *Th*) was deposited at the Museum of the Pharmacognosy Department, Faculty of Pharmacy, Cairo University.

### 3.2. Chemicals and Reagents

All chemicals for phytochemical analysis were from Sigma-Aldrich (Merck, Kenilworth, NJ, USA). Chemicals for the in vivo assays were obtained as follows: streptozotocin (STZ) and glucose (Sigma-Aldrich, Merck, USA), tween 80 (El-Gomhoria, Cairo, Egypt), daonil™, glibenclamide (Sanofi Co., Cairo, Egypt), cholesterol and HDL (BioChain, Eureka Dr, Newark, CA, USA), triglycerides (XpressBio, Wedgewood Blvd, Suite 103, Frederick, MD, USA), nitric oxide (StressGen, Glanford Ave., Suite 350, Victoria, British Colombia, Canada). Rat insulin ELISA (CUSABIO, Wuhan, China), glucose transporter 2 (GLUT2) ELISA (MyBioSource, Beijing, China), and malondialdehyde ELISA (LSBio, Fourth Avenue Suite 900. Seattle, USA) kits were used for the assay of biomarkers.

### 3.3. Preparation of the Crude Extracts

*Hs* and *Th* (1 Kg each) were separately macerated in 5 L of 70% (*v/v*) ethanol at room temperature (48 h), filtered then concentrated under reduced pressure at 40 °C to yield the ethanolic extracts as solid residues (110 and 100 g, respectively).

### 3.4. Phytochemical Analysis

#### 3.4.1. Acid Hydrolysis

Twenty-five grams of each extract were defatted with n-hexane (40–60 °C) and then subjected to the complete acid hydrolysis procedure (2N HCl, 100 °C, 2 h) [61]. The acidic solutions were extracted with ethyl acetate several times, affording an ethyl acetate extracts upon evaporation which were subjected to Sephadex LH-20 column using MeOH: H_2_O (1:1) then 100% MeOH, as eluents to attain pure aglycones. The major aglycones were identified by their typical ultraviolet (UV) absorbance using UV spectrophotometer (Shimadzu UV-240) and characteristic ^1^H-NMR signals (Jeol EX-500 spectrometer; JOEL Inc., Tokyo, Japan). On the other hand, the minors were aligned by co-chromatography with authentic samples (formerly isolated and identified by our research group, Department of Phytochemistry and Plant Systematics) using paper chromatography (PC) (Whatman Ltd., Maidstone, Kent, England) and solvent systems; 50% AcOH and BAW (*n*-BuOH-AcOH-H_2_O 4:1:5). To discriminate between the sugar moieties, the aqueous layer of each extract was carefully neutralized, then exposed to the PC elution using BBPW (benzene: *n*-BuOH: pyridine: H_2_O; 1:5:3:3) versus standard sugars (E. Merck, Darmstadt, Germany) [62].

#### 3.4.2. Sample Preparation, HPLC Profiling and MS Analyses

##### Sample Preparation

The lyophilized extracts of *Hs* and *Th* were prepared for HPLC profiling and UPLC-MS/MS analyses following the afore-described method [62].

##### HPLC Profiling

To profile *Hs* and *Th* extracts, HPLC system comprising a Waters 1525 Binary Pump with a 7725i Rheodyne injection port, a Kromega Solvent Degasser, Waters 996 Photodiode Array Detector, and a Luna polar Omega column (3.6 µm, 250 × 4.6 mm, Phenomenex^®^) was used. Gradient elution of analytes was carried out with acetonitrile; ACN (solvent A) and 0.1% triflouroacetic acid; TFA (solvent B) at a constant flow rate of 0.5 mL/min, with an injection volume of 5 μL. A non-linear gradient was applied: for the first 5 min, 90% B; for the next 10 min; 80% B, then 70 % B for 10 min, 65% B for additional 15 min, 50% B for 15 min, followed by 100% B for 7 min and finally 90% B for the last 3 min. The detection wavelengths were 210, 250, 285, and 375 nm.

##### UPLC–HRMS-MS Analysis

The HR-MS-MS analysis was carried out on MaXis 4G instrument (Bruker Daltonics^®^) coupled to an Ultimate 3000 HPLC (Thermo Fisher Scientific^®^). A UPLC-method was applied as follows: (with 0.1% formic acid in H_2_O as solvent A and 100% ACN as solvent B), an isocratic gradient of 10% B for 10 min, 10% to 100% B in 30 min, 100% B for an additional 10 min, using a flow rate of 0.3 mL/min; 5 μL injection volume and UV detector (UV/VIS) wavelength monitoring at 210, 254, 280, and 360 nm. The separation was carried out on a Nucleoshell 2.7 µm 150 × 2 mm column (Macherey-Nagel^®^), and the range for MS acquisition was *m/z* 50–1800 Daltons (Da).

A capillary voltage of 4500 V, nebulizer gas pressure (nitrogen) of 2 (1.6) bar, ion source temperature of 200 °C, the dry gas flow of 9 L/min source temperature, and spectral rates of 3 Hz for MS1 and 10 Hz for MS^2^ were used. For acquiring MS/MS fragmentation, the 10 most intense ions per MS^1^ were selected for subsequent CID with stepped CID energy applied. The employed parameters for tandem MS were applied as previously detailed [63].

##### Data Analysis

Raw data visualization was performed using Compass Data Analysis 4.4 (Bruker Daltonics^®^) while Metaboscape 3.0 (Bruker Daltonics^®^) was utilized for picking molecular features, and for raw data treatment and pre-processing. T-ReX 3D (Time aligned Region Complete eXtraction) algorithm was implemented for retention time alignment and the simultaneous detection and combination of isotopes, adducts, and fragments innate to the same compound into one feature. A bucket table was then generated containing all the detected features along with their retention time (Rt), measured *m/z*, molecular weight, and detected ions [64]. The cataloged ions table was created with an intensity threshold 10e3 and 10e4 for negative and positive ionization modes, respectively setting a retention time range from 1 to 40 min with restricted mass range *m/z* from 120 to 1800 Da.

##### Molecular Networking and Compounds Dereplication

The MS/MS (MS^2^) data (positive and negative modes) were independently uploaded as .mgf files to the publicly available Global Natural Product Social molecular networking (GNPS) platform (http://gnps.ucsd.edu, accession on: 31 March 2020) running the feature-based molecular networking online workflow [19].

The data was analyzed with a parent mass tolerance of 0.05 Da and an MS/MS fragment ion tolerance of 0.05 Da to create consensus spectra. A network was then made with a cosine score above 0.65 and more than 6 matched peaks between two consensus mass spectra to be connected with an edge. The spectra in the network were queried against GNPS’ spectral libraries (NIST13, MassBank, and Respect) following the same manner as the input parameters. The output molecular network was visualized and analyzed using Cytoscape (3.4.0) [65].

Manual putative structures identification was assisted by Sirius + CSI:FingerID 4.0.1 for the molecular formula prediction [21] and structural hits search with *m/z* tolerance set to 20 ppm connected to online Pubchem and DEREP-NP database, which was manually integrated [66].

### 3.5. Biological Assays

#### 3.5.1. In Vitro Anti-Diabetic Assays (Enzymes Inhibitory Assays)

The α-amylase bioassay method was adopted from [67] and modified as described by [63]. Similarly, the estimation of the *β*-glucosidase inhibitory activity was carried out following Sancheti’s procedures [68]. And finally, Conforti’s protocol [69] was used for the evaluation of pancreatic lipase inhibitory activity.

#### 3.5.2. In Vivo Antidiabetic Study

##### Experimental Animals

Thirty-six male albino Wistar rats (150–200 g) were randomly housed in cages in an air-conditioned animal room with 12 h dark and light cycles. Each cell (size 26 × 41 cm) lodged with four rats. The rats were kept for 24 h for acclimatization before the experiment. The rats were fed with a standard laboratory diet and with tap water ad libitum. All animal procedures were performed after approval by October University for Modern Sciences and Arts (MSA) Ethics Committee (BP1/EC1/2019PD).

##### Induction of Diabetes Mellitus

After 12 h of fasting, diabetes was induced by a single dose of intraperitoneal injection of freshly prepared STZ (50 mg/kg b.wt) in 0.1 M citrate buffer (pH 4.5). After 48 h, blood samples were collected, and the glucose levels were measured to confirm induction of diabetes; rats with serum glucose levels above 300 mg dL^−1^ were considered diabetic.

##### Experimental Design

After induction, rats were randomly divided into six groups (six rats/group). The diabetic control group received 1 mL of 1% tween 20, whereas glibenclamide group was treated with 6.5 mg/kg glibenclamide. Two treatment groups for each of *Th* and *Hs* extracts where doses of 100 mg kg^−1^ and 200 mg kg^−1^ orally administered for 21 days.

##### Preparation of Serum and Tissue Samples

At the end of the experimental period, all animals were deprived of food overnight and then sacrificed by cervical decapitation after being anaesthetized by ether inhalation.

By centrifugation of blood at 1000 rpm for 10 min and storing at −20 °C, serum was separated to assess the levels of glucose, insulin, cholesterol, HDL, triglycerides, and nitric oxide.

The liver and pancreas were excised immediately, rinsed with isotonic saline, and minced. Homogenates were prepared with 10% (*w/v*) phosphate-buffered saline (PBS, 0.1 mol/L, pH = 7.4). The supernatants of these homogenates, centrifuged at 10,000 rpm for 5 min at −4 °C, were directly used for the determination of pancreatic MDA and liver GLUT2.

##### Biochemical Analyses

Serum glucose, cholesterol, HDL, and triglycerides were determined spectrophotometrically using Beckman and Coulter AU480 chemical analyzer, however, nitric oxide was defined colorimetrically. Meanwhile, serum insulin, pancreatic MDA, and liver GLUT2 were assayed by ELISA using TECAN Spectra Classic plate reader (Crailsheim, Germany).

### 3.6. Statistical Analyses

All the in vitro assays were carried out in triplicate. The results were displayed as mean ± SD. The IC_50_ (concentration necessary for 50% inhibition of enzyme activity) was calculated by constructing a linear regression curve showing extracts concentrations (from 75 to 600 μg mL^−1^) for *α*-amylase, (from 60 to 600 μg mL^−1^) for *β*-glucosidase and (from 12 to 100 μg mL^−1^) for pancreatic lipase on the x-axis and percentage inhibition on the y-axis [70]. An unpaired student t-test was used for statistical comparison between the two groups. *p* < 0.05 is considered statistically significant. All analyses were done using the SPSS v22.0 (IBM, Chicago, IL, USA) and Graphpad Prism (6.01) (San Diego, CA, USA).

## 4. Conclusions

Molecular networking via the GNPS platform allowed the expansion of the metabolic profiles of the two Egyptian seagrasses; *H. stipulacea* and *T. hemprichii* with 144 metabolites. These metabolites were found to belong to different metabolic classes, five of them were tentatively identified for the first time from nature. Albeit, the genuineness of the tentatively assigned new metabolites needs to be further assessed through their isolation and full characterization through other spectroscopic techniques, i.e., NMR. Additionally, the created MNs inferred the clear prevalence of phenolics, flavonoids, and diterpenes as chemotaxonomic markers with the predominance of flavone acyl glycosides (acetyl and malonyl derivatives) and steviol glycosides in *H. stipulaceae*. In contrast, sulphated flavone glycosides and cinnamic esters of tartaric acid were found to be exclusive in *T. hemprichii*. Considering the deep interrogation of their chemical profiles and in comparison to that of *Enhalus* and other seagrasses, the three taxa were quite distinguished and confirmed their placement in three separate taxonomic positions. In addition, as antidiabetic candidates, both seagrasses showed potential activities. Treatment of diabetic rats with the low and high doses of *Hs* and *Th* extracts resulted in a significant reduction in serum glucose levels and a rise in insulin levels by 4-folds like glibenclamide treatment. Moreover, *Hs* extract demonstrated 9- and 13-fold increase in serum NO for 100 and 200 mg/kg/day doses, respectively, compared to diabetic controls. Their mechanism of action was likely envisioned through the improvement of glucose uptake by the tissues through the restoration of liver GLUT-2. Moreover, both extracts ameliorated oxidative stress status generated by the free radicals and dyslipidemia under the diabetic condition. *H. stipulaceae* extract showed a significantly higher potency in the treatment of diabetes and the associated oxidative stress and hyperlipidemia. These findings offer a promising consideration regarding *H. stipulaceae* into further natural antidiabetic-oriented studies.

## Figures and Tables

**Figure 1 marinedrugs-19-00279-f001:**
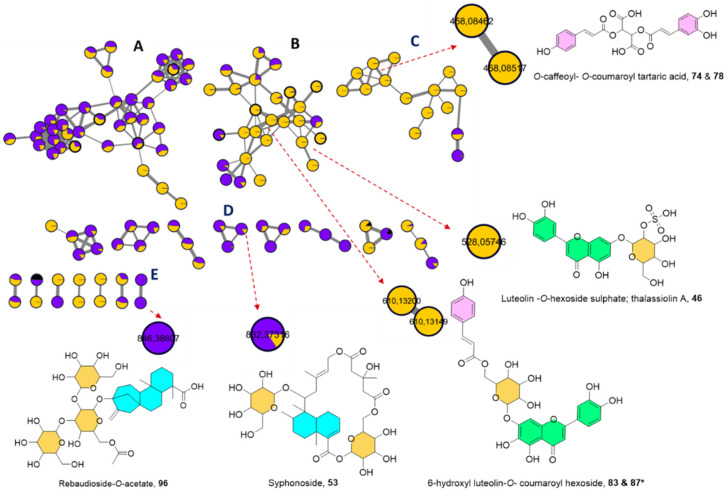
The enlarged negative molecular network created using MS/MS data (negative mode) from *Halophila stipulaceae* (purple nodes) and *Thalassia hempiricii* (*Th*) (yellow nodes). The network is displayed as a pie chart to reflect the relative abundance of each ion in both extracts. The black color corresponds to the solvent used as a blank.

**Figure 2 marinedrugs-19-00279-f002:**
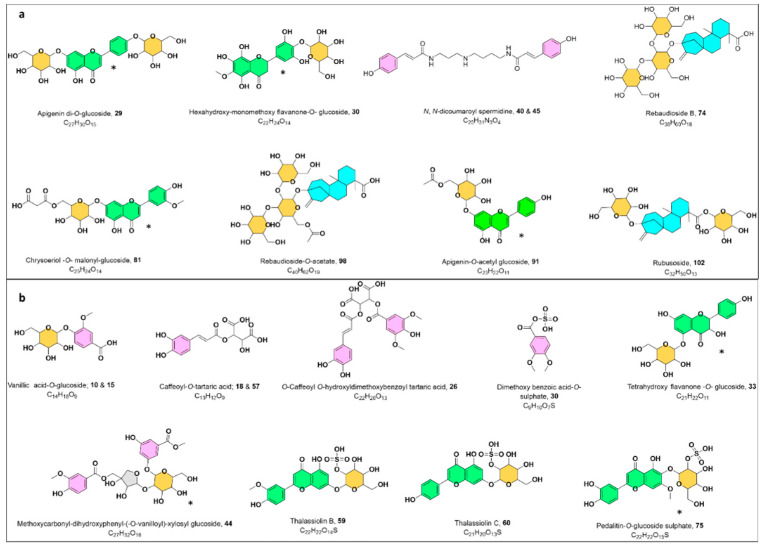
Some characteristic chemical structures; (**a**) *Halophila stipulaceae* (*Hs*), (**b**) *Thalassia hempiricii* (*Th*); * Position of the substitution may vary. Detailed information about the annotated metabolites (including the retention time, observed molecular ion, fragmentation pattern, molecular formula, and mass error) are displayed in Appendix A.

**Figure 3 marinedrugs-19-00279-f003:**
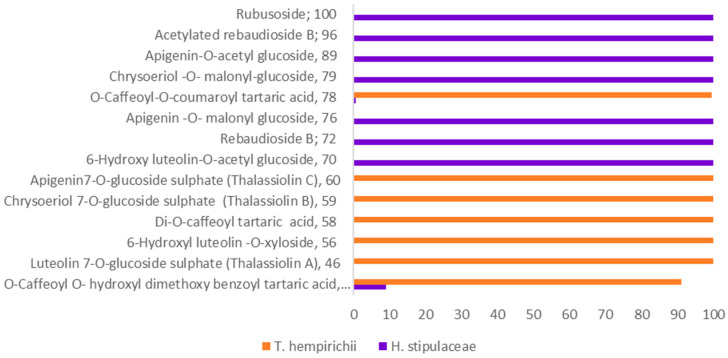
Distribution of the significant metabolites contributing to the chemotaxonomic significance of *H. stipulaceae* (*Hs*) and *T. hempirihii* (*Th*).

**Figure 4 marinedrugs-19-00279-f004:**
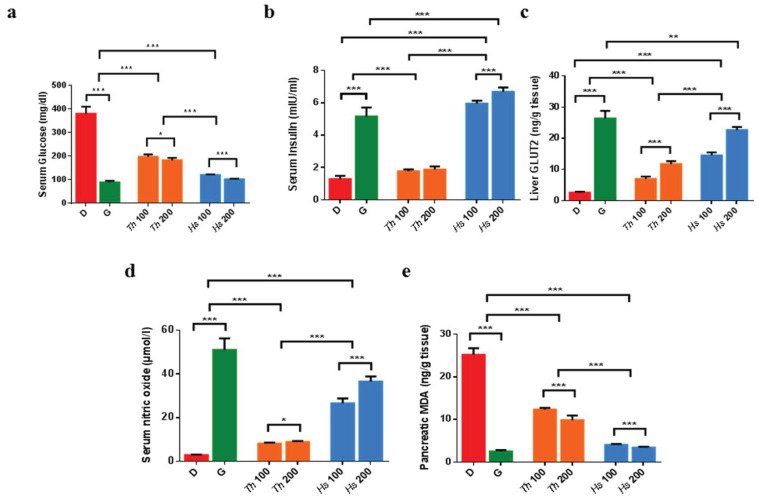
In vivo antidiabetic biomarkers in different studied groups; (**a**) Serum glucose level, (**b**) serum insulin level; (**c**) pancreatic GLUT-2 level; (**d**) Serum NO level; (**e**) pancreatic MDA level. Results are expressed as mean ± SD. D, diabetic control; G, glibenclamide 6.5 mg/kg/day; *Th* 100, *Th* 100 mg/kg/day; *Th* 200, *Th* 200 mg/kg/day; *Hs* 100, *Hs* 100 mg/kg/day; *Hs* 200, *Hs* 200 mg/kg/day. * Significant difference at *p* < 0.5, **: Significant difference at *p* < 0.01, ***: Significant difference at *p* < 0.001.

## Data Availability

For both investigated organisms, the positive MN is available at: https://gnps.ucsd.edu/ProteoSAFe/status.jsp?task=755e87f25be44b35830001c8e7e6421d accessed date: 30 September 2020, and the negative MN at: https://gnps.ucsd.edu/ProteoSAFe/status.jsp?task=590c967b10834ffd8b1bda94f37e67e3, accessed date: 30 September 2020.

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
