# Peer review of "Molecular Networking Leveraging the Secondary Metabolomes Space of Halophila stipulaceae (Forsk.) Aschers. and Thalassia hemprichii (Ehrenb. ex Solms) Asch. in Tandem with Their Chemosystematics and Antidiabetic Potentials"

_marinedrugs, 2021, doi:10.3390/md19050279_

Round 1

Reviewer 1 Report

Dear Authors,

This work describe the metabolomics study of  two seagrasses  Thalassia hemprichii and Halophila stipulacea via GNPS molecular networking. Five compounds were identified as potentially new.  Furthermore, both the Thalassia hemprichii and Halophila stipulacea crude extracts showed potential antidiabetic activities. 

1) Page 3, lines 101-102, "were additionally confirmed via their UV and 1H-NMR spectral data (Supplmentary Table S1). " but these 1H NMR were not found in supplementary data.

2) Page 13, lines 562-564, the links for resulting GNPS molecular networks were not working. 

3) Compounds 16, 26, 30, and 96 were claimed new due to the additional of methoxy and acetoxy units but these unit could be arise from using of methanol and ethyl acetate solvents, thus could be artifacts/unnatural products. How do Authors exclude this possibility?

4) Purification and collecting NMR measurements of these new compounds will further confirm their proposed structures.

Author Response

Dear Reviewer,

First thanks for handling our manuscript (marinedrugs-1213609: Molecular Networking Leveraging the Secondary Metabolomes Space of Halophila stipulaceae (Forsk.) Aschers. and Thalassia hemprichii (Ehrenb. ex Solms) Asch. in Tandem with Their Chemosystematics and Antidiabetic Potentials). We are grateful for the reviewer’s diligent effort to help improve our manuscript. Provided attached is a detailed rebuttal list addressing all comments. 

Please, kindly find enclosed the response to the reviewer's comments point by point file. We hope the current version is better and deserves to be acceptable for publishing now.

Reviewer 2 Report

The manuscript "Molecular Networking Leveraging the Secondary Metabolomes Space of Halophila stipulaceae (Forsk.) Aschers. and Thalassia 
hemprichii (Ehrenb. ex Solms) Asch. in Tandem with Their Chemosystematics and Antidiabetic Potentials" by Nesrine et al, has presented a good research story but in its present form, this manuscript can not be accepted. I suggest the following changes or points which need to be addressed before this manuscript will consider for publication in marine drugs.

  1. Line 29: UPLC-HRMS/MS, kindly provide the abbreviation
  2. Line 31: ....encompassing phenolic acids, flavonoids, terpenoids, and lipids...., molecular networking is not classifying the group of compounds.,,,,, reframe this sentence with proper meaning
  3. Line 56: scientific names should be in italics
  4. Line 123-125: author mentioned about 190 singletons, but it's missing in the figure. How the clustering was done? which annotation programme was used? did the authors use derepliration? Kindly provide the methodological details in the method section
  5. Figure no 2: Please provide the mass/error for observed and predicted  values, for all the compounds u left at her home
  6. Line 478-480: how about sampling permission number or any such kind of permission required for this research study. 
  7. Section 3.4.2, 3.4.4, 3.4.6 can be combined together.
  8. Line 535: ....MS1 and 10 Hz for MS2...10Hz is too high? please rectify 
  9. Line 562 - 564: provide the links separately in the data availability section
  10. In supplementary file: Fig S3, both base peak chromatogram seems to me in negative mode. Please check and rectify
  11. Fig S4, both base peak chromatogram seems to me in positive mode. Please check and rectify

Author Response

Dear Reviewer,

First thanks for handling our manuscript (marinedrugs-1213609: Molecular Networking Leveraging the Secondary Metabolomes Space of Halophila stipulaceae (Forsk.) Aschers. and Thalassia hemprichii (Ehrenb. ex Solms) Asch. in Tandem with Their Chemosystematics and Antidiabetic Potentials). We are grateful for the diligent effort to help improve our manuscript. Provided attached is a detailed rebuttal list addressing all comments. 

Please, kindly find enclosed the response to the reviewer's comments point by point file. We hope the current version is better and deserves to be acceptable for publishing now.

Round 2

Reviewer 1 Report

Dear Authors,

Thank you for addressing the comments positively. 

1) Page 6, line 223, [M-H] should be [M-H]-

2) Page 6, line 217, [M-H]- should be [M-H]-

3) Page 6, line 229, di-O-coumaroyl should be di-O-coumaroyl.

4) Page 10, lines 424-425, stated phenolics can be a chemotaxonomical marker of Enhalus, Halophila, and Thalassia. But phenolics can be found in many different organisms, normally a specific type of chemical skeleton in terpenes, alkaloids, or phenolics was considered as chemotaxonomical marker.

Author Response

Dear Reviewer,

Thanks again for handling our manuscript (marinedrugs-1213609: Molecular Networking Leveraging the Secondary Metabolomes Space of Halophila stipulaceae (Forsk.) Aschers. and Thalassia hemprichii (Ehrenb. ex

Solms) Asch. in Tandem with Their Chemosystematics and Antidiabetic Potentials). We are grateful for your diligent effort to help improve our manuscript. Provided below is a detailed rebuttal list addressing all comments, 

We hope the current version is better and deserves to be acceptable for publishing now.

  • Thank you for addressing the comments positively. 

You are most welcome. 

  • Page 6, line 223, [M-H] should be [M-H]-

Revised as recommended and highlighted within the text (line 223).

  • Page 6, line 217, [M-H]- should be [M-H]-

Revised as recommended and highlighted within the text (line 217).

  • Page 6, line 229, di-O-coumaroyl should be di-O-coumaroyl.

Revised as recommended and highlighted within the text (line 230).

  • Page 10, lines 424-425, stated phenolics can be a chemotaxonomical marker of EnhalusHalophila, and Thalassia. But phenolics can be found in many different organisms, normally a specific type of chemical skeleton in terpenes, alkaloids, or phenolics was considered as chemotaxonomical marker.

Among the different secondary metabolites, phenolics have been widely used and effective in chemotaxonomy for a number of reasons. In the current research, it has been shown that the phenolic profiles of the three genera are being exploited as useful chemotaxonomic markers. This is due to their extensive varieties as derivatives of benzoic acids, cinnamic acid esters, and flavonoids as discussed in the manuscript.

The sentences (line 423-426) within the text were slightly modified to be clearer and understandable and highlighted to be:

Decisively, the phenolic entities, i.e., the extensive varieties as derivatives of benzoic acids, cinnamic acid esters, and flavonoids, of Enhalus, Halophila, and Thalassia could be efficient chemotaxonomic markers advocating their placement in three different taxonomic ranks.

Reviewer 2 Report

After the revision by authors, the quality of this manuscript has improved immensely. Thus, I recommend this manuscript to be published in it's present form. 

Author Response

Dear Reviewer,

Thank you so much for your positive comments. On behalf of all the co-authors, we would like to thank you again, and we are highly appreciating your feedback and comments.